# An Effective Sublingual Vaccine, MV140, Safely Reduces Risk of Recurrent Urinary Tract Infection in Women

**DOI:** 10.3390/pathogens12030359

**Published:** 2023-02-21

**Authors:** J. Curtis Nickel, R. Christopher Doiron

**Affiliations:** Department of Urology, Queen’s University, Kingston, ON K7L 3N6, Canada

**Keywords:** urinary tract infections, vaccine, recurrent urinary tract infections, women’s health

## Abstract

Uncomplicated recurrent urinary tract infections (rUTIs) in women are associated with episodic bothersome symptoms and have a significant impact on the mental and physical quality of life. Treatment with antibiotics (short- and long-term dosing) results in acute and chronic side effects and costs and promotes general antibiotic resistance. Improved nonantibiotic management of rUTI in women represents a true, unmet medical need. MV140 is a novel sublingual mucosal-based bacterial vaccine developed for the prevention of rUTI in women. Based on observational, prospective, and randomized placebo-controlled studies, MV140 has been shown to safely prevent (or reduce the risk of) UTIs, reduce antibiotic use, overall management costs, and patient burden while improving the overall quality of life in women suffering from rUTIs.

## 1. The Problem of Recurrent Urinary Tract Infection

Uncomplicated urinary tract infections (UTIs) are one of the most common infections in women [1], yet the consensus among many in the medical community appears to be that they are not serious infections, do not lead to dangerous short- or long-term outcomes, and can be simply managed with antibiotic therapy. However, most clinicians are realizing that this is not the case for recurrent UTIs [2]. Of the 11% of women who will develop a simple uncomplicated UTI each year, 25% will experience another UTI within the next 6 months [3,4]. Recurrent UTI (rUTI), defined as three or more UTIs in 12 months (or two or more in 6 months) [5] has a yearly incidence of 3% [3]. Women suffering from rUTI experience a significant impact on both their mental and physical quality of life. They report a higher risk of depression, stress, anxiety as well as sexual dysfunction and physical disabilities, resulting in time off work and from important daily activities [6,7,8]. Additionally, most women with rUTI experience short- and long-term side effects from antibiotics, which lead to a further deterioration of their health status [9]. While most are familiar with the common side effects associated with short courses of antibiotics (allergies, GI symptoms, etc.), taking long-term prophylactic antibiotics for up to 6 months or longer can lead to serious (neurologic and GI) and even life-threatening side effects [10,11]. Women with rUTI have been documented to be understandably frustrated and even angry at the medical profession for how they manage this disease [8]. This has led women to not trust the medical profession to look after them [12], and for them to seek alternatives that may not be safe or effective. Then, there is the specter of rising antimicrobial resistance, an acknowledged major worldwide problem [13]. Such antimicrobial resistance can lead to management difficulties in patients taking many and varied antibiotic doses or prolonged prophylaxis but it also affects the treatment of all infectious diseases as our antibiotic pipeline dries up.

## 2. Alternatives to Antibiotics

International rUTI guidelines [5,14,15] have recommended some alternatives to antibiotics to reduce the risk of further infections, some with moderate evidence and some with poor evidence [16,17]. The best evidence approaches from these recommendations include the use of vaginal estrogen in postmenopausal women, cranberry extract supplements, and increasing water intake. Less evidence-based suggestions include D-Mannose supplements, intravesical hyaluronic acid, and probiotics. One recommendation that is available in Europe [15] but not North America [5,14], likely because of availability and approval reasons, is the use of immune modulation therapy (e.g., vaccines). Meta-analyses of vaccine trials for the management of rUTIs [18] indicate that overall, vaccines provide risk reduction. Of note, was the observation that two trials comparing one vaccine, MV140, to prophylactic antibiotic therapy, statistically pushed the analysis into a positive impact [18].

## 3. MV140 Sublingual Vaccine for Prevention of rUTI in Women

MV140 (Uromune^®^, Inmunotek S.L., Spain) consists of a suspension of whole-cell heat-inactivated bacteria (300 Formazin Turbidity Units) in glycerol, sodium chloride, artificial pineapple flavoring, and water. Included are equal percentages of selected strains of four bacterial species (V121 *Escherichia coli*, V113 *Klebsiella pneumoniae*, V125 *Enterococcus faecalis*, and V127 *Proteus vulgaris*) (Table 1). MV140 is administered daily sublingually by spraying two sprays of 100 µL each, under the tongue for 3 months.

## 4. Mechanism of Action of MV140 Sublingual Vaccine for rUTI

MV140, a polyvalent bacterial whole-cell-based sublingual vaccine has been developed for the prevention of UTIs and is currently in prelicensed phase 2–3 development, available under named patient (special access) programs in 26 countries. The beneficial role of bacterial preparations for the prevention of rUTIs has been extensively evaluated [18,19,20]. Only recently have we gained some understanding of the mechanisms of action of most of these vaccines to induce protective immunity in the bladder that mediates this protection [20,21,22]. Mechanistic studies have shown that sublingual MV140 induces antibody production [23] and activates human dendritic cells to generate T helper (Th) 1, Th17, and interleukin-10, producing anti-inflammatory T-cell responses in secondary lymphoid organs and locally in the bladder [24]. The induction of adaptive immunity likely underlies the clinical protection observed following treatment discontinuation, although trained immunity (activation of the innate immune cells may result in enhanced responsiveness to subsequent bacterial triggers) could also play a role [23,25]. See Figure 1.

## 5. Nonrandomized and Observational Studies of MV140

A recent systematic review [26] identified 73 studies examining MV140 and UTI, 19 of which examined its use in the prevention of recurrent UTIs. Five of those met criteria to critically review its role in the prevention of uncomplicated rUTIs in women (standardized definition of rUTI, female, at least one outcome parameter to include UTI-free rate after vaccination). The analysis reported higher UTI-free rates among those women treated with the vaccine daily for 3 months (35%–58%) versus 6 months of antibiotic prophylaxis (0%) in two comparative studies [27,28]. The first study included 319 women (159 treated with MV140) [27], while the second study includes 669 women (360 treated with MV140) [28]. Three observational uncontrolled studies involving women with rUTIs [29,30,31] showed UTI-free rates in treated subjects ranging from 33%–78% over 9–24 months of follow-up. A total of 1400 women with rUTIs were evaluated in these five studies [27,28,29,30,31] (Table 2).

## 6. Early Canadian Clinical Experience Study

The preliminary interim analysis of the first 25 subjects (pre-COVID-19 cohort) enrolled in a Health Canada-approved first-in-North America clinical experience study [33] was comparable to the five previous studies described in the previous section. Briefly, the reduction in UTI rate was 82% for the 9 months postvaccination (mean reduction in total cohort to 2.1 UTIs/month), compared to the prevaccination UTI rate (mean 11.5 UTIs/month). The UTI-free rate for the 9 months postvaccination was 48% (12/25). At 12 months of follow-up, 80% of subjects (20/25) reported they had moderately/markedly improved compared to prevaccination. This first-in-North America clinical trial has been completed (the last patient completed September 2022), the database is locked, and the results have been analyzed and are expected mid-2023).

## 7. Effectiveness of MV140 in Other Populations of Patients with an rUTI

Several studies have enrolled men, children, the frail elderly, or subjects with complicated UTIs. In studies examining the impact of MV140 on men with rUTIs, one [30] enrolled 136 men who experienced a 30% UTI-free rate at a 6-month follow-up, while another smaller study [34] described 14 men with rUTIs who reported a UTI-free rate of 71% at 12 months (10/14). Other studies evaluating subjects (men and/or women) with complicated UTIs including men with prostatitis [35], neurogenic bladder [36], autoimmune disease and treatment-mediated immunosuppressed patients [37], chronic renal disease and kidney transplant [38,39,40,41], lymphoproliferative disorders [42], frail elderly [43,44], children [45] and postsurgery [46] consistently reported favorable UTI-free rates ranging from 30–50% and significant UTI reduction rates and/or improved quality of life after MV140 treatment.

Strong evidence examining the use of MV140 in one particularly vulnerable population, the frail elderly, where the clinical impact would be enormous, is limited but certainly suggestive of effectiveness. In one prospective analysis of MV140 in the frail elderly [43] from nursing homes, 200 subjects (160 females, mean age 82.67 years; 40 males, mean age 80.23 years) had a median of 4.0 UTI (or asymptomatic bacteriuria) episodes per month treated with antibiotics. The UTI rate decreased significantly following MV140 treatment in females to a median of 0.1 UTI/month (range 0.0–0.4 UTIs/month). UTI-free rates for the 12 months following initiation of the vaccine were observed in 18% of women, while 81.7% were treated for <3 UTIs/year.

## 8. Cost-Effectiveness of MV140

Clinical studies described in this review confirm that MV140 significantly reduces the number of UTIs when compared to prevaccination UTI rates, antibiotic prophylaxis, and placebo. It is logical that a reduction in UTI risk would translate into decreased healthcare utilization (fewer trips to seek medical attention, fewer urine cultures, etc.,) and antibiotic cost. A recent quasi-experimental, pretest–post-test, single-center study [47] including 166 women with rUTI, vaccinated with MV140 in real-life clinical practice, prospectively assessed healthcare utilization and associated costs. Primary care physician and emergency room visits, urine cultures, ultrasound exams (and other ancillary testing), and antibiotic costs significantly decreased compared to prevaccination (all *p* < 0.001). This resulted in an over 50% reduction in healthcare expenditure per patient/year. Reducing the risk of rUTI in women being managed for rUTI does indeed lead to considerable cost savings. This does not take into account the very real impact that reducing antibiotic consumption in this population could theoretically have on rapidly increasing antibiotic resistance rates. Reduction in UTI rates associated with less antibiotic use should reduce the risk of antibiotic resistance, not only in individual patients or the entire rUTI cohort but also in the general population.

## 9. Pivotal MV140 Randomized Placebo-Controlled Clinical Trial

In a recently published European multicenter, randomized, double-blind, placebo-controlled parallel-group one-year trial [32], (NCT02543827) 240 women with rUTIs were allocated to receive MV140 for 3 or 6 months, or placebo for 6 months, in a 1:1:1 ratio. The primary endpoint was the number of UTIs in the 9-month study period following 3 months of intervention. Key secondary endpoints were the percentage of women UTI-free over the above period and the time to UTI onset. In this pivotal study, MV140, either for 3- or 6-month administration, significantly decreased the median number of UTI episodes from a median of 3.0 to 0.0 compared to placebo in the 9-month efficacy period (i.e., following 3 months of intervention) [32]. A significant increase in the UTI-free rate of over 2-fold was found, being 55.7% and 58.0% in subjects receiving MV140 for 3 or 6 months, respectively, compared to 25.0% in the placebo group. The median time until the appearance of the first UTI after 3 months of treatment was 275.0 days [IQR, 87.0–275.0] in both the MV140 3-month and 6-month groups compared to 48.0 days in the placebo group.

A subanalysis of the RCT examining the effect of MV140 on patient burden [48], included the evaluation of relevant secondary analyses such as the impact on patient safety, symptom severity, antibiotic use, and multiple aspects of quality of life (SF-36). Vaccinated groups experienced significantly less overall UTI symptoms, fewer days on antibiotics, and significantly improved total, general, and physical (SF-36) quality of life improvements. By safely reducing the risk of UTI, MV140 significantly reduces the personal burden of UTI disease in women suffering from the health consequences of rUTIs [48].

## 10. MV140 Safety

The studies evaluating MV140 in populations of women with rUTIs have not reported any major safety concerns. There were no adverse reactions (ARs) reported for the two major comparative studies comparing MV140 to antibiotics [27,28]. One serious adverse reaction (SAR) (allergic reaction) and seven minor ARs (post-nasal drip, stinging around the mouth, pruritus over old BCG scar, pruritus over the abdomen, intermittent abdominal pain, mild nausea, and exacerbation of underlying asthma) in the UK prospective study [29]. Other minor side effects included in one study of 784 subjects [30], were dry mouth in 8, gastritis in 4, and general illness in 3, while in another study of 166 subjects [31], minor side effects included two reports of mild glossitis and one flare-up of rheumatoid arthritis (which was not believed to be associated with the treatment). Only 2 out of 1407 women treated with MV140 in the five studies selected as part of the systematic review decided to discontinue the treatment. Our preliminary report of the first-in-North American clinical experience study [33] confirmed safety data, with few adverse events (AEs) observed: five nonserious AEs and one SAE noted among 25 subjects studied: only one mild and self-limited AE was potentially related to the vaccine.

The overall safety of MV140 is further confirmed by safety reports from over 22,000 subjects receiving the vaccine (data collected until December 2021) in compassionate or named patient programs (ClinicalTrial.Gov: NCT 04173013). Only 15 reports of ARs have been filed for over 1.5 million doses (data on file, Pharmacovigilance Department, Inmunotek, Spain).

The safety issues regarding the use of MV140 in a population of women with rUTI were carefully and prospectively documented in the published report of the randomized placebo-controlled trial [32]. There was a total of 81 AEs in the placebo group and 76 and 48 in MV140 3-month and 6-month groups, respectively. The most common AEs (≥5% of participants) were chest infections, candidiasis, and vaginitis. The seven SAEs reported in five participants were assessed as not unexpected or logically related to MV140. Only 9 out of the 205 AEs reported in the trial were considered as adverse reactions to the study intervention, presenting in a total of five subjects (2.2%): two from placebo (2.6%), three from MV140 3-month group (3.9%) and none from MV140 6-month group. MV140 appears to be a very safe intervention.

## 11. Future Considerations

MV140 is presently not available in the US or Canada. It has been available since 2010 in clinical practice in different countries worldwide on a named patient product (NPP) basis or similar special access or compassionate use, including Spain, Portugal, the United Kingdom, Lithuania, the Netherlands, Sweden, Norway, Australia, New Zealand, and Chile. Recently, MV140 has been approved in Mexico and Dominican Republic and submitted to Health Canada for registration. Studies in tentative planning stages include evaluating the efficacy and safety in the elderly residing in long-term care homes, in children with rUTIs, and adult patients with complicated rUTIs (e.g., catheterized patients and/or patients with neurogenic bladder). Subject to further assessment is the impact of the repeated administration of MV140 following the potential return of the rUTI health state and its possible combination with vaccines for associated infections [22].

## 12. Conclusions

The improved nonantibiotic management of rUTIs in women represents a true, unmet medical need. It appears the sublingual vaccine MV140 safely prevents (or reduces the risk of) UTI, and reduces antibiotic use, overall management costs, and patient burden while improving the overall quality of life in women suffering from rUTIs.

## Figures and Tables

**Figure 1 pathogens-12-00359-f001:**
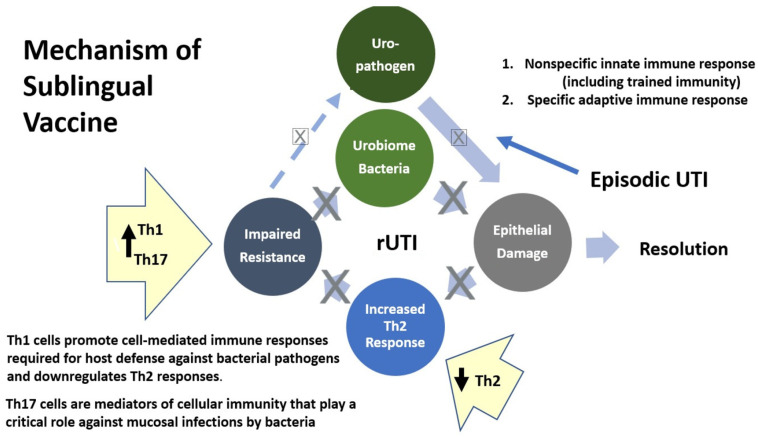
Proposed mechanism of action for MV140: An episodic UTI from a uropathogen entering the urinary tract causes an acute infection with inflammation that typically resolves with antibiotic therapy. In some cases, in susceptible women, the epithelial damage results in an increase in T helper 2 (Th2) response, which impairs resistance to both exogenous bacteria (from outside the urinary tract) and endogenous bacteria (within the patient’s urinary microbiome) causing a scenario of recurrent UTI. MV140 increases Th1 and Th17 cells which both downregulate the abnormal Th2 response and promote or mediate cellular immune responses, resulting in a decreased risk of rUTI from bacteria outside and inside the bladder microbiome.

**Table 1 pathogens-12-00359-t001:** Composition and administration of MV140 mucosal vaccine for rUTIs.

Composition
Four whole-cell inactivated bacteria *Escherichia coli* (25%) *Klebsiella pneumoniae* (25%) *Proteus vulgaris* (25%) *Enterococcus faecalis* (25%)
Administration
Sublingual route Self-administered 2 sprays under the tongue once daily 3-month treatment

**Table 2 pathogens-12-00359-t002:** Description of trials evaluating the efficacy of MV140 in women with rUTIs. All included trials must have included a UTI-free outcome (percentage of subjects with no UTI after vaccination).

Study Design (Reference)	Subjects (rUTI in Women)	Treatment	Efficacy Findings
Randomized placebo-controlled Lorenzo-Gomez et al [32]	78	Placebo 6 months	9-month UTI reduction (after 3-month treatment): median 3.0 placebo vs. 0.0 MV140 groups (*p* < 0.001) 9-month UTI-free rate 55.7–58.0% MV140 vs. 25.0% placebo groups (*p* < 0.001)
	77	MV140 3 months (+Placebo 3 months)	
	75	MV140 6 months	
Retrospective antibiotic-comparator Lorenzo-Gomez et al [27]	159	MV140 3 months	15-month UTI-free rate: 34.6% MV140 vs. 0% Antibiotic group (*p* < 0.001) 15-month UTI reduction: mean 5.75 Antibiotics vs. 1.35 MV140 (*p* < 0.001)
	160	TMP/SMX 6 months	
Retrospective antibiotic-comparator Lorenzo-Gomez et al. [28]	360	MV140 3 months	12-month UTI-free rate: 90.3% MV140 vs. 0% antibiotic group (*p* < 0.001) Delayed UTI onset postvaccination: median 180 days MV140 vs. 19 days ATB group (*p* < 0.001)
	339	TMP/SMX or nitrofurantoin 6 months	
Prospective observational descriptive noncomparative Yang et al [29]	75	MV140 3 months	12-month UTI-free rate: 78.7%
Prospective observational descriptive noncomparative Ramirez-Sevilla et al [30]	648	MV140 3 months	6-month UTI-free rate (postvaccination): 32.3% Rate of 0–1 UTI (6 months postvaccination): 65.9%
Prospective observational longitudinal Carrion-Lopez et al [31]	166	MV140 3 months	12-month/24-month UTI-free rate: 52.4%/44.5% UTI reduction/year: 54.6% (*p* < 0.001 compared to prevaccination)
Prospective observational real-world early clinical experience—preliminary report Nickel et al [33]	25 (pre-COVID-19 cohort)	MV140 3 months	9-month UTI-free rate (postvaccination): 48% UTI/month reduction: 82% (compared to the year prevaccination) Self-reported moderate/marked improvement: 80% of subjects
Prospective observational real-world early clinical experience—Final report Nickel et al. (this work)	64	MV140 3 months	9-month UTI-free rate (postvaccination): Pending 2023 UTI/month reduction: Pending 2023 Self-reported moderate/marked improvement: Pending 2023

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
