# Peer review of "An Effective Sublingual Vaccine, MV140, Safely Reduces Risk of Recurrent Urinary Tract Infection in Women"

_pathogens, 2023, doi:10.3390/pathogens12030359_

Round 1

Reviewer 1 Report

This is a nicely written short review on the efficacy and safety of the MV140 Uromune vaccine against recurrent urinary tract infection. I provide a few comments below to improve its clarity. (It would have helped the review process for the manuscript to have line numbers to refer to.):

  1. The MV140 vaccine is frequently referred to in the literature by the manufacturer’s trade name “Uromune”, and yet that name is never used in this article. The authors should make it clear at the start of section 3, when the vaccine is introduced, that MV140 is also called Uromune - this may help the reader to connect with previous literature on the vaccine. 

  2. In section 1, paragraph 1, the sentence starting “For the 11% of women…” is incomplete. What is the second half of the sentence (“... 25% within the next 6 months.”) referring to?

  3. Table 1 is inefficiently presented, and probably unnecessary, as the same information is provided in the text. Also, the descriptions of administration are not consistent: “twice daily for 3 months” (in the text) is not the same as “2 sprays once daily” (in the table). Just get rid of Table 1.

  4. The concept of “trained immunity” is not widely known or understood - it may be worth a brief description in section 5, particularly since if true, it would further enhance the value of MV140.  

  5. Table 2 is confusingly presented, including undefined acronyms (for example: ATB), divergent strategies for assessing efficacy, items bolded for no apparent reason, and everything jammed into a compact space in a way that makes it difficult to read and interpret. As with Table 1, some of the content of Table 2 is also redundant with the text. If the authors want to keep Table 2, they should present it more clearly, and remove text in Section 6 that provides redundant information (e.g. the number of patients in the trials). The text should simply provide a high level overview or summary of key points from the table.

  6. Section 7 reiterates data in Table 2 as well. If the authors want to specifically highlight that study, then take it out of Table 2, and just deal with it in the text of Section 7.

  7. Section 10 is again a description of data presented in Table 2. It isn’t necessary to do both - if the authors want to highlight this data, don’t put it in Table 2. (Maybe Table 2 could just be removed?) Also, the information in Section 10 should probably be moved to earlier - couldn’t it just immediately follow section 6?

  8. I found it disappointing that there was no effort to compare MV140 with alternative UTI vaccines currently being tested. Perhaps that was beyond the scope of this review, but other reviews of efforts to generate a vaccine against recurrent UTI have noted several vaccine products being tested. With regard to this review, it is notable that MV140 seems to be more effective in trials to date than any of the alternatives. 

Author Response

Reply to reviewer #1.  All revisions are tracked in the revised manuscript.

  1.  The trade name for MV140 trade name in Europe, Mexico and and many other countries is "Uromune" however that name will not be the trade name in North America (Health Canada and likely FDA will not agree to the that specific trade name).  However, I have indicated the trade name "Uromune" the first time MV140 is mentioned in the Introduction section.
  2. The reviewer is correct.  We have revised that sentence accordingly.
  3. The description of vaccine application has been expanded in the text and the a minor revision in the table.  The authors believe that it is helpful to have this summary table to better illustrate the composition and administration of the vaccine.  The manuscript is short to start with so an extra table should not be a burden.  However if the editor totally agrees with the reviewer, we are OK with the editor deleting Table 1 (this is not our preference).
  4. An short explanation of "trained immunity" has been added to the section.
  5.   Agree with reviewer that "ATB" is not a good abbreviation. It is now spelled out as "Antibiotic" .  Regarding the table, we would like to respectfully disagree with the reviewer.  For many such review articles, such summary tables are the backbone of the manuscript and are used by other readers as the main reference point for previous studies (rather than the text).  The manuscript is very short and it should not be a problem to leave the Table intact.
  6. Same reply as #5
  7. Same reply as #5
  8. The authors agree with the reviewer that the additional discussion of other vaccine strategies is outside the scope of this article.  The authors also totally agree with the reviewer that an article focusing on such a comparison would be useful.

Reviewer 2 Report

Dear Authors,

The work “An effective sublingual vaccine, MV140, safely reduces risk of recurrent urinary tract infection in women” (pathogens-2204919)  is interesting and noteworthy.

The manuscript is dedicated to MV140, a sublingual bacterial vaccine of whole-cell heat-inactivated bacteria of selected strains of four bacterial species Escherichia coli, Klebsiella pneumoniae, Enterococcus faecalis, and Proteus vulgaris. The vaccine is indicated for the prevention of recurrent urinary tract infections.

The authors reviewed many publications, including some of their own, indicating the beneficial effect of this vaccine in reducing the recurrence of urinary tract infections, mainly in women with recurrent urinary tract infections, but also in men and children.

Clinical studies described in this review confirm that MV140 significantly reduces the number of urinary tract infections compared to pre-vaccination urinary tract infection rate, and compared to antibiotic treatment.

Figure 1 is very interesting and helps to better understand the proposed mechanism of operation of the V140.

What is very important, the use of MV140 can contribute to the reduction of antibiotic resistance among bacteria that cause urinary tract infections, which will certainly translate into a significant reduction in the costs associated with the treatment of these infections.

The manuscript can be published after introducing slight corrections:

·   Table 1 does not look like a table - generally, it could be removed, because the description is present in the text.

·   what does the abbreviation AEs mean? – no explanation in the text (point 11, page 6)

· the bibliography should be improved and adapted to Pathogens journal guidelines.

Best regards,

Author Response

Reply to Reviewer #2 suggested corrections. The revisions are tracked in the revised manuscript.

  1. Table 1 has been reformatted to look more like a typical Table.  The authors believe that this summary table would be useful for the readers.  The manuscript is short so there should be no problem with length of article by leaving this table in place.
  2. AE has been spelled out as adverse event in the relevant text.

The authors would like to thank the reviewer for the kind comments regarding our article.